# RAC1B Suppresses TGF-β1-Dependent Cell Migration in Pancreatic Carcinoma Cells through Inhibition of the TGF-β Type I Receptor ALK5

**DOI:** 10.3390/cancers11050691

**Published:** 2019-05-17

**Authors:** Hendrik Ungefroren, Hannah Otterbein, Christian Fiedler, Koichiro Mihara, Morley D. Hollenberg, Frank Gieseler, Hendrik Lehnert, David Witte

**Affiliations:** 1First Department of Medicine, University Hospital Schleswig-Holstein, Campus Lübeck, D-23538 Lübeck, Germany; hendrik.ungefroren@uksh.de (H.U.); hannahotterbein@web.de (H.O.); c.fiedler93@gmx.de (C.F.); hendrik.lehnert@uni-luebeck.de (H.L.); 2Clinic for General Surgery, Visceral, Thoracic, Transplantation and Pediatric Surgery, University Hospital Schleswig-Holstein, Campus Kiel, D-24105 Kiel, Germany; 3Departments of Physiology and Pharmacology and Medicine, Inflammation Research Network, Snyder Institute for Chronic Diseases, University of Calgary, Cumming School of Medicine, Calgary, AB T2N 4N1, Canada; mihara@ucalgary.ca (K.M.); mhollenb@ucalgary.ca (M.D.H.); 4Department of Oncology, University Hospital Schleswig-Holstein, Campus Lübeck, D-23538 Lübeck, Germany; frank.gieseler@uksh.de

**Keywords:** RAC1B, ALK5, cell migration, pancreatic carcinoma, RNA interference, CRISPR/Cas9, TGF-β

## Abstract

The small GTPase Ras-related C3 botulinum toxin substrate 1B (RAC1B) has been shown previously by RNA interference-mediated knockdown (KD) to function as a powerful inhibitor of transforming growth factor (TGF)-β1-induced cell migration and epithelial-mesenchymal transition in epithelial cells, but the underlying mechanism has remained enigmatic. Using pancreatic carcinoma cells, we show that both KD and Clustered Regularly Interspaced Short Palindromic Repeats (CRISPR)/Cas9-mediated knockout (KO) of RAC1B increased the expression of the TGF-β type I receptor ALK5 (activin receptor-like kinase 5), but this effect was more pronounced in CRISPR-KO cells. Of note, in KO, but not KD cells, ALK5 upregulation was associated with resensitization of *TGFBR1* to induction by TGF-β1 stimulation. RAC1B KO also increased TGF-β1-induced C-terminal SMAD3 phosphorylation, SMAD3 transcriptional activity, growth inhibition, and cell migration. The KD of ALK5 expression by RNA interference or inactivation of the ALK5 kinase activity by dominant-negative interference or ATP-competitive inhibition rescued the cells from the RAC1B KD/KO-mediated increase in TGF-β1-induced cell migration, whereas the ectopic expression of kinase-active ALK5 mimicked this RAC1B KD/KO effect. We conclude that RAC1B downregulates the abundance of ALK5 and SMAD3 signaling, thereby attenuating TGF-β/SMAD3-driven cellular responses, such as growth inhibition and cell motility.

## 1. Introduction

RAC1B, and its more prominent isoform, Ras-related C3 botulinum toxin substrate 1 (RAC1), are encoded by the human *RAC1* gene. RAC1B differs from RAC1 by in-frame insertion of exon 3b, encoding for 19 amino acids, resulting in a small GTPase with impaired enzymatic activity but an accelerated ability to exchange GDP to GTP [1]. RAC1B can promote cell cycle progression and survival; however, its role in other processes driving tumor progression like epithelial-mesenchymal transition (EMT), cell motility, and metastasis is less well understood. The inclusion of exon 3b in the RAC1B isoform results in alterations in signaling properties and cellular functions of RAC1B (reviewed in [1]), some of which are antagonistic to that of RAC1. For instance, our RNAi-triggered knockdown (KD) analyses suggest that endogenous RAC1B and RAC1 suppress and promote, respectively, TGF-β1-dependent migration (chemokinesis) of “normal” and malignant pancreatic epithelial cells [2,3], as well as carcinoma-derived cell lines of the breast [4,5] and prostate (H.U., unpublished data). In addition, our published data suggest that RAC1B suppression of cell migration may involve downregulation of TGF-β1-induced phosphorylation of SMAD3C [2], p38 MAPK (microtubule-associated protein kinase), and extracellular signal-regulated kinase (ERK)1/2 MAPK [3], which are critical for TGF-β1-induced migration. However, the mechanism(s) whereby RAC1B interferes with SMAD and MAPK activation are not known yet. TGF-β ligand-induced stimulation of TGF-β type I receptor activin receptor-like kinase 5 (ALK5) promotes the phosphorylation-activation of SMAD3, p38 MAPK, and ERK MAPK, thus suggesting that RAC1B may downregulate the expression of ALK5 or its kinase activity to inhibit these downstream targets. In the current study, we investigated the functional significance of RAC1B-mediated reduction of ALK5 abundance on TGF-β1-stimulated cell migration, using the pancreatic ductal adenocarcinoma (PDAC)-derived cell lines Panc1 and Colo357.

## 2. Results

### 2.1. Knockout (KO) and Knockdown (KD) of RAC1B Increased Expression of ALK5

Previous data obtained with Panc1 cells have shown that KD of RAC1B via a siRNA targeting exon 3b of *RAC1* resulted in elevated levels of ALK5 mRNA [3]. To confirm the RNA interference-based results and to be able to study TGF-β1-dependent cellular responses in a RAC1B-null background, we generated Panc1 cells in which exon 3b of *RAC1* was deleted by Clustered Regularly Interspaced Short Palindromic Repeats (CRISPR)/Cas9 technology (Panc1-RAC1B-KO). RAC1B, unlike the related RAC1, was undetectable in these cells at the mRNA level, as measured by quantitative real-time RT-PCR (qPCR), and protein level, as assessed by immunoblot analysis (Appendix A). In contrast, Panc1-RAC1B-KD cells maintained residual expression of endogenous RAC1B protein (19 ± 15% of control) 48 h after transfection (Appendix A). To reveal whether a complete lack of RAC1B reproduces the KD effect on ALK5 expression and eventually sensitizes *TGFBR1* to TGF-β1 stimulation, we measured ALK5 expression in Panc1-RAC1B-KO cells. Panc1-RAC1B-KD and KO cells were stimulated or not with TGF-β1 for 24 h and subjected to qPCR and immunoblot analysis for ALK5. Intriguingly, ALK5 mRNA expression under basal conditions (non-TGF-β1 treated) was enhanced in Panc1-RAC1B-KD (Figure 1A) and RAC1B-KO cells (Figure 1B), but this enhancement was much more pronounced in the KO cells (Figure 1B). Likewise, TGF-β1 treatment for 24 h failed to increase ALK5 mRNA levels significantly in both control siRNA-transfected cells (Figure 1A) and in CRISPR/Cas9-engineered vector control cells (Figure 1B). However, upon downmodulation of RAC1B, TGF-β1 was able to increase ALK5 mRNA abundance further, either marginally in KD cells (Figure 1A) or strongly in KO cells (Figure 1B). At the protein level, RAC1B KD alone (without TGF-β1 stimulation) resulted in a 1.6-fold increase in ALK5 protein abundance, but there was no statistically significant increase after 12 h or 24 h of TGF-β1 treatment (Figure 1A). In contrast, a strong induction of ALK5 abundance after 24 h of TGF-β1 treatment was evident in Panc1-RAC1B-KO but not in vector control cells (Figure 1B). Both ALK5 mRNA and protein levels were also elevated in Colo357 cells following RAC1B KD (Appendix A). From these data, it can be concluded that RAC1B negatively controls TGF-β1-dependent mRNA and protein abundance of ALK5.

### 2.2. Stable KO of RAC1B Enhanced TGF-β1-Induced SMAD3 Activation and SMAD3-Dependent Responses

Ligand activation of ALK5 initiates SMAD signaling, and this is crucial for TGF-β1-induced cellular responses, including cell migration/invasion [2]. We, therefore, addressed the question of whether the increase in ALK5 abundance of RAC1B-depleted cells translates into enhanced C-terminal phosphorylation (reflecting activation) of SMAD3 and SMAD3-dependent responses. Intriguingly, the ratio of C-terminally phosphorylated SMAD3 (p-SMAD3C) to total SMAD3 protein was greatly enhanced in KO cells when compared to vector control cells and reached peak values at 0.5–1 h after the addition of TGF-β1 (Figure 2A). The time-course analysis revealed that levels of phospho-SMAD3C remained well above the control even after 24 h of treatment (Figure 2A). This result suggests that RAC1B is a powerful inhibitor of TGF-β1-induced SMAD3 activation.

To evaluate the functional consequences of the high p-SMAD3C/SMAD3 ratio for general TGF-β/SMAD-mediated transcription, we performed reporter gene assays with the SMAD3-responsive reporter gene p(CAGA)_12_-luc. Interestingly, this reporter responded to TGF-β1 treatment in KO cells with a dramatic induction in luciferase activity compared to vector controls (Figure 2B).

Growth inhibition represents a prominent SMAD3-dependent response to TGF-β1 [6]. Given the high SMAD3C activation in KO cells, we hypothesized that the antiproliferative effect of TGF-β1 is more pronounced in these cells compared to the vector control cells. To this end, cell counting experiments indicated that the growth inhibitory effect of TGF-β1 was much stronger in Panc1-RAC1B-KO cells than in the vector controls (Figure 2C), suggesting that RAC1B acts as an inhibitor of TGF-β1-induced growth arrest.

Taken together, the RAC1B depletion-induced upregulation of ALK5 enhanced TGF-β1-induced SMAD3 activation and SMAD3 transcriptional responses.

### 2.3. Stable KO of RAC1B Enhanced TGF-β1-Dependent Migration

Previously, we have shown that KD of RAC1B results in an upregulation of TGF-β1-induced migratory (chemokinetic) activity in PDAC and breast cancer cells [2,3,4]. To reveal whether RAC1B KO can duplicate this effect, we measured cell migration in real-time in Panc1-RAC1B-KO cells. Consistent with results of Panc1-RAC1B-KD cells, we found that RAC1B KO increased TGF-β1-induced migration as compared to the vector control cells (Figure 3A, magenta curve/tracing D vs. green curve/tracing B). A quantitative comparison of peak TGF-β1-dependent migratory activity resulted in cell index (CI) values of 1.5 ± 0.4 for Panc1-vector controls and 2.4 ± 0.7 (*p* = 0.02, student’s *t*-test) for Panc1-RAC1B-KO cells (Figure 3A). Next, we compared TGF-β1-induced migration of Panc1 cells in which RAC1B was knocked down vs. knocked out. Based on the residual expression of endogenous RAC1B protein in Panc1-RAC1B-KD cells (see Appendix A), we reasoned that maximal migratory activity should be higher in cells in which RAC1B was deleted. Consistent with this expectation, we found that KO cells exhibited a significantly higher migration index than KD cells (Figure 3B, magenta curve/tracing B vs. green curve/tracing A). The quantitative analysis of peak TGF-β1-dependent migration yielded CI values of 1.5 ± 0.4 for Panc1-RAC1B-KD and 2.4 ± 0.7 (*p* = 0.02) for Panc1-RAC1B-KO cells (Figure 3B). These data suggest that RAC1B might act in a concentration-dependent manner to dampen the sensitivity of these cells to TGF-β1-induced migration.

### 2.4. Downregulation of the Protein Abundance of ALK5 Suppressed TGF-β1-Induced Cell Migration Increase by RAC1B Deletion

Above, we have shown that RAC1B KD or KO strongly enhanced the TGF-β1 effect on cell motility. To demonstrate the crucial role of ALK5 in mediating TGF-β1-induced chemokinesis, we knocked down ALK5 by transient transfection of wild-type Panc1 cells and Colo357 cells with ALK5 siRNA. In these cells, the TGF-β1 effect on migration was strongly reduced (Appendix A). To obtain evidence that the upregulation of ALK5 is also responsible for the RAC1B KD/KO-mediated increase in TGF-β1-dependent cell migration, we cotransfected Panc1 or Colo357 cells with RAC1B and ALK5 siRNA, while Panc1-RAC1B-KO cells were transfected with ALK5 siRNA alone. Again, the high migratory activity afforded by RAC1B KD/KO was strongly reduced upon ALK5 KD in both Panc1 (Figure 4) and Colo357 (Appendix A) cells. These data show that the increase in ALK5 expression accounted at least in part for the elevated TGF-β1-induced chemokinesis in RAC1B depleted cells.

### 2.5. ALK5 Kinase Activity is Required for RAC1B KD-Mediated Increase in TGF-β1-Induced Cell Migration

In an earlier study, we showed that a kinase-dead ALK5 mutant (ALK5-K232R), which acts in a dominant negative fashion to suppress ALK5 kinase activity, prevented upregulation of TGF-β-induced phospho-p38 by RAC1B KD [3]. Since both ALK5 kinase activity and p38 activation are required for TGF-β1-induced migration, we reasoned that ectopic ALK5-KR expression should also attenuate the RAC1B KD/KO-dependent rise in migratory activity. To evaluate this possibility, we employed Panc1-RAC1B-KO cells transiently transfected with ALK5-K232R. Remarkably, the TGF-β1-induced increase in migration afforded by RAC1B suppression was strongly reduced by kinase-inactive ALK5 (Figure 5A). Very similar data were obtained with Colo357-RAC1B-KD cells treated with the ALK5 kinase inhibitor SB431542 (Appendix A).

Finally, if the increase in ALK5 kinase activity (associated with the enhanced protein expression) is responsible for the RAC1B KD/KO-mediated rise in TGF-β1-dependent cell migration, then the ectopic expression of a kinase-active ALK5 in Panc1 cells should be capable of mimicking this effect. Strikingly, expression of ALK5-T204D, a constitutively active ALK5 mutant [7] (without concomitant KD or KO of RAC1B), was able to increase the migratory activity of Panc1 cells strongly over that of empty vector control cells (Figure 5B). The data presented in Figure 4; Figure 5 clearly demonstrate that the increase in ALK5 abundance and kinase function is not simply an epiphenomenon but is causally involved in the negative control of TGF-β1-dependent cell migration by RAC1B.

## 3. Discussion

In previous reports using various benign and malignant cells from pancreas and breast tissues, we have demonstrated negative regulation of TGF-β1-dependent cellular responses, such as EMT and cell migration/chemokinesis by RAC1B [2,3,4,5]. Here, we sought to unravel the mechanistic basis of this unexpected suppressive role of RAC1B. Previously, we observed increased mRNA expression of ALK5 in Panc1-RAC1B-KD cells, suggesting the possibility that elevated expression of ALK5 and the associated increase in kinase activity are responsible for increased sensitivity of cells to TGF-β1-mediated responses. This idea was also supported by previous findings indicating increased basal and ligand-dependent phosphorylation of the ALK5 substrates SMAD3 and p38 MAPK upon RAC1B KD [2,3]. To this end, we observed that transient KD of RAC1B by RNA interference in Panc1 or Colo357 cells or stable genomic deletion of exon 3b of *RAC1* by CRISPR/Cas9 technology in Panc1 cells resulted in an upregulation of the mRNA and protein abundance of ALK5. The increase in the mRNA/protein abundance of ALK5 was more pronounced in RAC1B KO than in KD cells. Moreover, while no further TGF-β1-induced increase in the protein abundance of ALK5 (12 and 24 h stimulation) was detected in Panc1-RAC1B-KD cells, a strong increase in protein abundance of ALK5 by TGF-β1 stimulation was observed in RAC1B-KO cells, supporting the notion that a complete loss of RAC1B in cells sensitized *TGFBR1* to TGF-β1 stimulation. To reveal whether the dramatic increase in ALK5 abundance in RAC1B KO vs. vector control cells was associated with enhanced signaling by TGF-β1, we monitored the activation of the SMAD pathway and various SMAD-dependent responses. Remarkably, the ratio of phosphorylated SMAD3C to total SMAD3—reflecting the state of SMAD3 activation—was much greater in KO cells compared to controls. Moreover, SMAD3C phosphorylation was prolonged in RAC1B-KO cells compared to vector control cells and remained elevated even after 24 h of TGF-β1 treatment. Likewise, RAC1B-KO cells responded to TGF-β1 with dramatically increased transcriptional activity of the SMAD3-responsive reporter p(CAGA)_12_-luc, and both responses even exceeded those observed earlier in KD cells [2]. Last, we observed that KO cells responded to TGF-β1 with greatly increased growth inhibition. A comparison of the chemokinetic properties of Panc1-RAC1B-KO and vector control cells in response to TGF-β1 stimulation revealed that these were higher in the KO cells (see Figure 3A). When quantitatively comparing the peak migratory response to TGF-β1 of RAC1B-KD cells (which still express ~19% of the protein abundance of endogenous RAC1B) and that of RAC1B KO cells (with undetectable mRNA/protein abundance of RAC1B), we observed a significantly higher migration rate in KO vs. KD cells (see Figure 3B). These data suggest that RAC1B negatively controls, in a dose-dependent manner, ALK5 expression and, as a consequence, SMAD3 activation and SMAD3-dependent cellular responses.

Since the possibility remained that ALK5 suppression by RAC1B was only an epiphenomenon rather than being causally involved in negative regulation of TGF-β1 signaling, we sought to prove the functional role of ALK5 using TGF-β1-induced cell migration as a read-out. We demonstrate that RAC1B suppresses both ALK5 mRNA and protein abundance to inhibit TGF-β-induced chemokinesis and that the associated decrease in ALK5 kinase activity is critical in this respect. This is reinforced by the demonstration that ectopic expression of kinase-dead ALK5 in Panc1-RAC1B-KO cells (see Figure 5A) or SB431542 treatment of Colo357-RAC1B-KD cells (see Appendix A) reversed the RAC1B KO/KD-induced increase in migratory activity, while the overexpression of kinase-active ALK5 in Panc1 cells obviated the need for the addition of TGF-β ligand and/or the depletion of RAC1B to promote cell migration in these cells (see Figure 5B).

Since the abundance of the TGF-β receptors, including ALK5, closely correlates with TGF-β signaling and hence cellular responsiveness, RAC1B-mediated inhibition of ALK5 expression and function has important implications for TGF- β biology [8]. Specifically, ligand-dependent assembly of the activated TGF-β type II-ALK5 receptor complex initiates the TGF-β signaling pathway and sets the threshold for triggering SMAD and non-SMAD (i.e., p38) signaling pathways. The findings of the current study, therefore, are consistent with previous results indicating that KD of endogenous SMAD4 or SMAD3 abrogates the increase in TGF-β-induced migration of Panc1 and Colo357 cells in which RAC1B was depleted [2]. The quantitative changes in SMAD3C phosphorylation, SMAD3 transcriptional activity, and growth inhibition may be due to an increase in the protein abundance and hence kinase activity of ALK5, an assumption that is supported by the ability of ectopically expressed kinase-dead ALK5 (KR mutant) to inhibit RAC1B KD-induced hyperphosphorylation of p38 in Panc1 cells [3]. However, the prolonged activation of SMAD3 and p38 cannot be explained simply by changes in ALK5 abundance but rather may involve subsequent alterations of downstream signaling. In this context, it is interesting to note that we have preliminary data to indicate that RAC1B stimulates expression of SMAD7 (H.U., unpublished observation). SMAD7 is a key negative regulator of TGF-β signaling that inhibits TGF-β signaling by binding to ALK5 and blocking SMAD2/3 phosphorylation directly, by competing for receptor binding, or indirectly, by promoting ubiquitin-mediated degradation of ALK5 [9].

We observed that RAC1B suppresses ALK5 expression at the mRNA and protein level. Regarding the molecular mechanism, a decrease in transcriptional activity of *TGFBR1* does not seem to be responsible for the RAC1B KD effect on ALK5 as assessed by reporter gene assays with a *TGFBR1* promoter-reporter construct (H.U., unpublished observation). This result suggests that downregulation of the mRNA is the result of a posttranscriptional mechanism involving mRNA stability rather than an impact on de novo transcription. For instance, control of mRNA half-life by microRNAs targeting the 3′-UTR is a prominent mode of ALK5 regulation [10,11]. Whatever the precise mode of action is, we have identified here the small GTPase RAC1B as a powerful inhibitor of endogenous TGF-β signaling in pancreatic tumor cells.

## 4. Material and Methods

### 4.1. Antibodies and Reagents

The following primary antibodies were used: anti-Smad3, #ab40854, Abcam (Cambridge, UK), anti-HSP90, #sc-7947 and #sc-13119, and anti-TGF-β receptor I (V22), #sc-398, Santa Cruz Biotechnology (Heidelberg, Germany), anti-RAC1B, #09-271, Merck Millipore (Darmstadt, Germany), anti-RAC1, #610650, BD Transduction Laboratories (Heidelberg, Germany), and anti-phospho-Smad3(Ser423/425), #9514, and anti-GAPDH (14C10), #2118, Cell Signaling Technology (Frankfurt am Main, Germany). Horseradish peroxidase (HRP)-linked anti-rabbit, #7074, and anti-mouse, #7076, secondary antibodies were from Cell Signaling Technology. Recombinant human TGF-β1, #300-023, was provided by ReliaTech (Wolfenbüttel, Germany), and the ALK5 inhibitor SB431542 from Merck Millipore.

### 4.2. Cells

The Panc1 and Colo357 human PDAC-derived cell lines were originally obtained from ATCC (Manassas, VA, USA) and cultivated in RPMI 1640 supplemented with 10% fetal bovine serum, 1% Penicillin-Streptomycin-Glutamine (Life Technologies/Thermo Fisher Scientific, Schwerte, Germany) and 1% sodium pyruvate (Merck Millipore). The generation of Panc1 cell clones ectopically expressing ALK5-T204D was described in detail earlier [3,7]. These cells were stably transduced using the retroviral vector TJBA5bMoLink-neo [7], followed by a selection of transduced cells with geneticin (700 µg/mL, Life Technologies/Thermo Fisher Scientific) and generation of individual cell clones by limited dilution. Pooled empty vector transductants served as a control. Cell counting of cells detached with trypsin/EDTA was performed with a Neubauer chamber. Cell viability was confirmed by trypan blue exclusion.

### 4.3. Generation of RAC1 Exon 3b-Deleted Panc1 Cells by CRISPR/Cas9 Technology

The basic procedure of CRISPR genome KO was described previously [12]. We used online CRISPR guide design tool [13] to design optimum guide sequences for KO of *RAC1* exon 3b. Two sequences each of 5′ and 3′ genomic region flanking exon 3b were chosen (Appendix A). The forward and reverse guide sequences with cloning flanking sequences were synthesized by the University of Calgary Core DNA facilities. The annealed oligonucleotides were inserted in LentiCRISPR v2 (Addgene, Watertown, MA, USA). A mixture of 4 guide sequences or just lentiviral vector was transfected in Panc1 cells with Lipofectamine 3000 (Thermo Fisher Scientific, Waltham, MA, USA) and the transfected cells were selected with 1 μg/mL of puromycin. Since bystander effects and growth-suppressive pressure of the RAC1B KO prevented the formation of a pure population of RAC1B KO cells, we performed limited dilution of transfectants with increased concentrations of puromycin (up to 20 μg/mL) to generate a genetically homogenous population of RAC1B KO cells. The purity of these cells was verified by qPCR with primers selectively recognizing RAC1B (Appendix A).

### 4.4. QPCR Analysis

Total RNA was extracted from Panc1 cells using PeqGold RNAPure from Peqlab (Erlangen, Germany) and purified according to the manufacturer’s instructions. For each sample, 2.5 μg RNA was subjected to reverse transcription for 1 h at 37 °C, using 200 U M-MLV Reverse Transcriptase and 2.5 μM random hexamers (Life Technologies/Thermo Fisher Scientific) in a total volume of 20 μL. Relative mRNA expression of target genes was quantified by qPCR on an I-Cycler (BioRad) using Maxima SYBR Green Mastermix (Thermo Fisher Scientific). Data were normalized to the expression of either TATA-box-binding protein (TBP) or β-actin. For sequences of PCR primers, see Appendix A.

### 4.5. Transient Transfection of siRNAs and Plasmid Vectors

On day 1 after seeding into plates (Nunclon^TM^ Delta Surface) from Nunc (Roskilde, Denmark), the cells were transfected twice on two consecutive days with 25–50 nM of siRNA specific for RAC1B, scrambled control [2,3], or a pool of three different and validated siRNAs to ALK5 (Validated Stealth RNAi^TM^ siRNA (Set of 3) or matched Stealth RNAi^TM^ negative control (Life Technologies)) for 4 h using Lipofectamine 2000 for Panc1 cells and Lipofectamine RNAiMAX for Colo357 cells (both from Life Technologies) at a concentration of 0.5%. Additional validation of these siRNAs was done for RAC1B, RAC1 [2,3,4], and ALK5 [14,15]. Plasmid vectors (ALK5-K232R-HASL in pcDNA3, and pcDNA3) were transfected into Panc1 cells using Lipofectamine 2000 according to the manufacturers’ instructions.

### 4.6. Cell Lysis and Immunoblotting

Confluent cells were washed once with ice-cold PBS and lysed with 1 × PhosphoSafe lysis buffer (Merck Millipore). Following sonication and clearing, the total protein concentration of the supernatants was determined with the BioRad DC Protein Assay. Samples were subjected to gel electrophoresis using BioRad mini-PROTEAN TGX any-kD precast gels and blotted to 0.45 μm PVDF membranes. Membranes were blocked with nonfat dry milk or BSA and incubated with primary antibodies at 4 °C overnight. HRP-linked secondary antibodies and Amersham ECL Prime Detection Reagent (GE Healthcare, Munich, Germany) were used for chemiluminescent detection of proteins on a BioRad ChemiDoc XRS imaging system.

### 4.7. Dual Luciferase Assays

This assay was performed as described in detail earlier [3]. Briefly, Panc1-vector and RAC1B-KO cells were seeded in 96-well plates on day 1 and on day 2 transfected with p(CAGA)_12_-luc along with the Renilla luciferase-encoding vector pRL-TK-luc (Promega, Heidelberg, Germany) using Lipofectamine 2000 (Life Technologies). On day 3, cells were treated with 5 ng/mL TGF-β1 for 24 h, followed by lysis and dual luciferase measurement.

### 4.8. Real-Time Cell Migration Assays

The xCELLigence^®^ DP system (ACEA Biosciences, San Diego, CA, USA, distributed by OLS, Bremen, Germany) was employed for recording random cell migration of Panc1 cells. CIM plates-16 were prepared according to the instruction manual and previous descriptions [3,14]. The underside of the upper chambers of the CIM plate-16 was coated with 30 μL of collagen I (400 μg/mL). In all assays, RPMI with 1% fetal bovine serum was present in both the upper and lower chambers of each well of a CIM plate-16. The upper chamber of each well was loaded with 50,000–60,000 cells immediately after the addition of 5 ng/mL TGF-β1 to the cell suspensions. Data acquisition was done at intervals of 15, 30, or 60 min and analyzed with RTCA software (ACEA).

### 4.9. Statistical Analysis

Statistical significance was calculated using the unpaired two-tailed student’s *t*-test. Results were considered significant at *p* < 0.05 (*). Higher levels of significance were *p* < 0.01 (**) and *p* < 0.001 (***).

## 5. Conclusions

The data obtained in this study lend further support to the contention that negative control of TGF-β-dependent cell motility by RAC1B is exerted at the level of ALK5. The RAC1B-mediated suppression of ALK5 expression results in decreased ALK5 kinase activity, SMAD3 signaling, and SMAD3-dependent responses like growth inhibition and cell migration. Therefore, strategies to increase the expression and/or activity of RAC1B, or its generation by alternative splicing from *RAC1* [1] may be worth testing in cancers in which TGF-β1 drives invasion and metastatic dissemination [1].

## Figures and Tables

**Figure 1 cancers-11-00691-f001:**
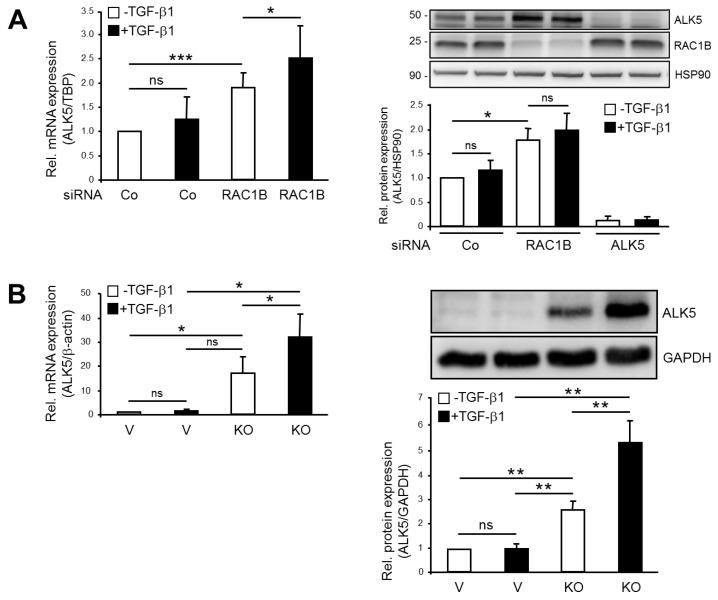
Effect of RAC1B knockdown (KD) and knockout (KO) on activin receptor-like kinase 5 (ALK5) expression in Panc1 cells. (**A**) Panc1-RAC1B-KD cells were generated by transfecting Panc1 cells twice with 50 nM of either control (Co) siRNA or RAC1B siRNA. Forty-eight hours after the second transfection, the cells were treated with transforming growth factor (TGF)-β1 for 24 h and subsequently processed for qPCR of ALK5 and TATA-box-binding protein (TBP) (left-hand panel) or immunoblot analysis of ALK5, RAC1B, and heat shock protein (HSP)90 (right-hand panel). The graphs show quantification of ALK5 data normalized to those for TBP (ΔΔCt values from qPCR) or HSP90 (densitometric values from immunoblots) and represent the mean ±SD from three independent experiments. (**B**) Panc1-RAC1B-KO and vector control cells were treated with TGF-β1 for 24 h and processed for either qPCR of ALK5 and β-actin (left-hand panel) or immunoblot analysis of ALK5 and Glyceraldehyde 3-phosphate dehydrogenase (GAPDH) (right-hand panel). Expression of RAC1B in these cells is shown in Appendix A. The graphs indicate the normalized data (mean ±SD, n = 3). The asterisks indicate significance (student’s *t*-test). V, vector control; ns, not significant.

**Figure 2 cancers-11-00691-f002:**
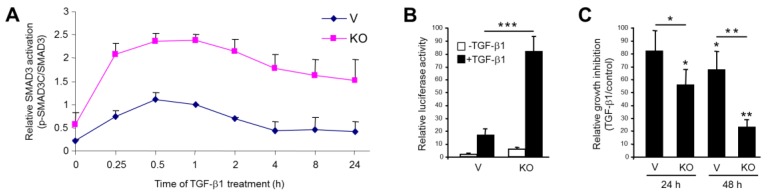
Effect of RAC1B knockout (KO) on transforming growth factor (TGF)-β1-induced phosphorylation of SMAD3C, SMAD3-dependent transcriptional activity, and growth inhibition in Panc1 cells. (**A**) Panc1-RAC1B-KO (KO) and vector control cells (V) were treated with TGF-β1 for various times (as indicated) and subjected to phospho-immunoblotting for SMAD3C and total SMAD3. Densitometric data for p-SMAD3C were normalized to those for total SMAD3 and represent the mean ± SD from three independent experiments. (**B**) Panc1-RAC1B-KO and vector control cells (V) were cotransfected with p(CAGA)_12_-luc and pRL-TK-luc, and after a 24 h treatment with TGF-β1, they were subjected to dual luciferase measurement. Data are the mean ±SD of six replicates per condition and are from one representative assay out of four assays performed in total with very similar results. (**C**) Panc1-RAC1B-KO and vector control cells were seeded at 50,000 cells per 12-well in normal growth medium and treated, or not, on the following day with TGF-β1 for 24 h or 48 h. Following detachment by trypsinization, viable cells were counted. Data are the mean ±SD of four wells processed in parallel and are displayed as % inhibition by TGF-β1 treatment relative to untreated control cells set arbitrarily to 100%. Data are representative of three experiments. The asterisks indicate significance (student’s *t*-test).

**Figure 3 cancers-11-00691-f003:**
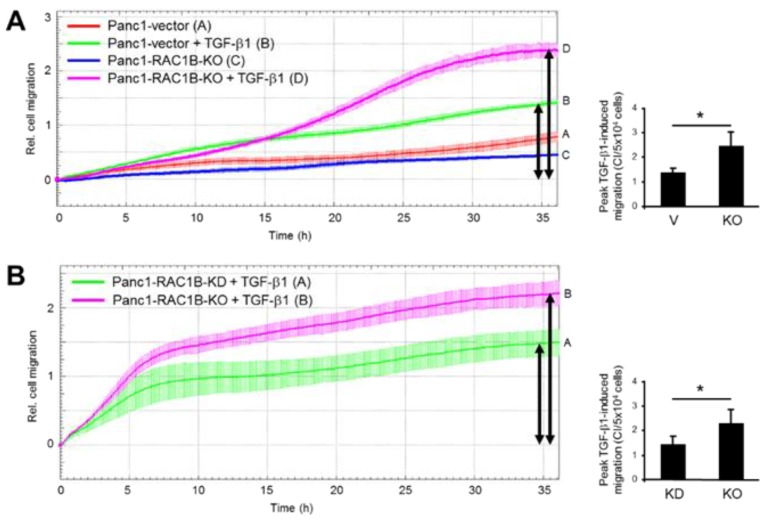
Transforming growth factor (TGF)-β1-induced migration in Panc1-RAC1B-knockout (KO) cells and knockdown (KD) cells. (**A**) Kinetics of TGF-β1-induced migration of Panc1-RAC1B-KO cells and corresponding vector control cells as measured by real-time cell migration assay. Shown is a representative assay, and data are the mean ± SD of three parallel wells. Differences between Panc1-RAC1B-KO + TGF-β1 (magenta curve, tracing D) and Panc1-vector + TGF-β1 (green curve, tracing B) are significant at 17:45 and all later time points. The bar graph shows quantification of peak TGF-β1-induced migratory activities (indicated by black arrows) of Panc1-vector vs. Panc1-RAC1B-KO cells by depicting the maximal cell index (CI) values (mean ± SD, n = 4). (**B**) Comparative analysis of TGF-β1-induced migration of Panc1-RAC1B-KD and KO cells by real-time cell migration assay. Panc1-RAC1B-KD were generated by transfecting cells twice with 50 nM of either control siRNA or RAC1B siRNA. Forty-eight hours after the second round of transfection, the cells were subjected together with Panc1-RAC1B-KO cells to real-time cell migration assay in the presence of 5 ng/mL TGF-β1. Data shown are the mean ± SD of three parallel wells and are significantly different at 05:15 and remain so at all later time points. The bar graph shows quantification of peak TGF-β1-induced migratory activities of Panc1-RAC1B-KD and KO cells by depicting their maximal CI values. Data represent the mean ± SD from four (KO) and eight (KD) assays. The asterisk indicates significance (student’s *t*-test).

**Figure 4 cancers-11-00691-f004:**
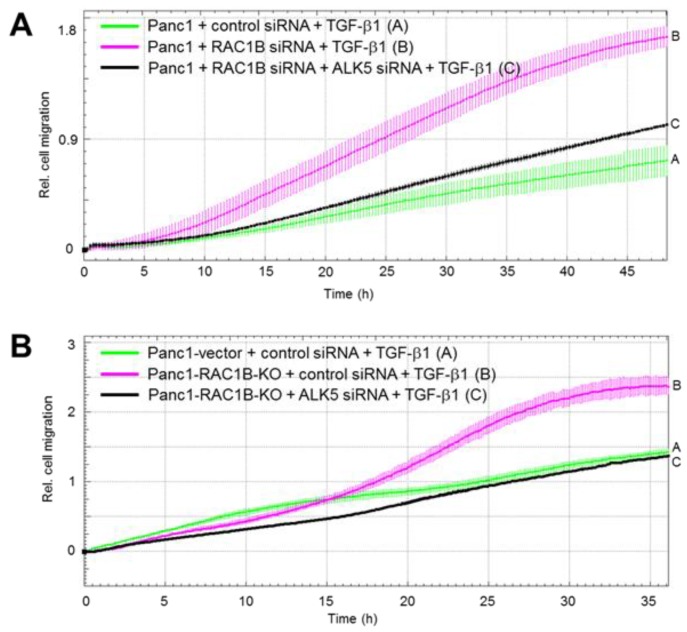
Effect of activin receptor-like kinase 5 (ALK5) knockdown (KD) on transforming growth factor (TGF)-β1-induced migration (chemokinesis) in Panc1-RAC1B-KD and knockout (KO) cells. (**A**) Panc1 cells were transfected twice with either 50 nM of control siRNA, 25 nM RAC1B siRNA+ 25 nM control siRNA, or 25 nM RAC1B siRNA + 25 nM ALK5 siRNA. Forty-eight hours after the second transfection, the cells were processed for migration assay on the xCELLigence platform. Immediately before the start of the assay, one-half of the cells received 5 ng/mL TGF-β1. (**B**) As in (**A**) except that Panc1-RAC1B-KO cells received 50 nM of either siCo or siALK5. Data in (**A**) and (**B**) are from one representative experiment and are the mean ±SD from 3–4 wells per condition. Differences between Panc1 + RAC1B siRNA + ALK5 siRNA + TGF-β1 (black curve, tracing C) and Panc1 + RAC1B siRNA + TGF-β1 (magenta curve, tracing B) are significant at 16:30 and all later time points, while in (**B**) those between Panc1-RAC1B-KO + ALK5 siRNA + TGF-β1 (black curve, tracing C) and Panc1-RAC1B-KO + control siRNA + TGF-β1 (magenta curve, tracing B) are significant at 07:00 and all later time points. Successful inhibition of RAC1B and ALK5 was verified by immunoblotting (data not shown). For functional validation of the ALK5 siRNA, see Figure 1A.

**Figure 5 cancers-11-00691-f005:**
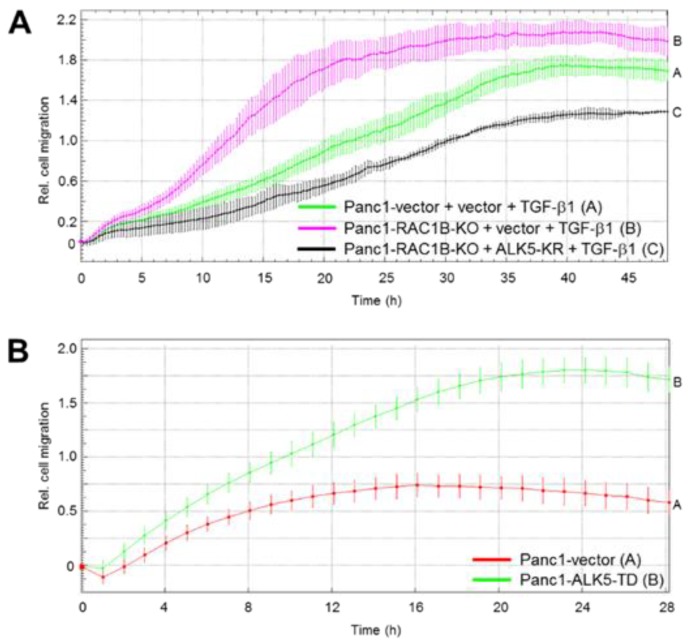
Effect of kinase-dead and kinase-active activin receptor-like kinase 5 (ALK5) on transforming growth factor (TGF)-β1-induced migration in Panc1 cells. (**A**) Panc1-RAC1B-knockout (KO) cells were transiently transfected with either empty vector or ALK5-K232R (ALK5-KR) and 48 h later subjected to real-time cell migration assay. Data are from one representative experiment out of three experiments performed in total and represent the mean ±SD from 3–4 wells per condition. Differences between Panc1-RAC1B-KO + ALK5-KR + TGF-β1 (black curve, tracing D) and Panc1-RAC1B-KO + vector + TGF-β1 (magenta curve, tracing B) are significant at 07:00 and all later time points. (**B**) Effect of kinase-active ALK5 on Panc1 cell migration. Panc1 cells stably expressing either empty vector or ALK5-T204D (ALK5-TD) were subjected to real-time cell migration assay. Data are from one representative experiment (three performed in total) and represent the mean ±SD from three parallel wells. Differences between Panc1-ALK5-TD (green curve, tracing B) and Panc1-vector (red curve, tracing A) are significant at 03:15 and all later time points.

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
