# Peer review of "RAC1B Suppresses TGF-β1-Dependent Cell Migration in Pancreatic Carcinoma Cells through Inhibition of the TGF-β Type I Receptor ALK5"

_cancers, 2019, doi:10.3390/cancers11050691_

Round 1
Reviewer 1 Report
The article is interesting, However the authors needs to address the comments below.
The authors in Figure 2B show only the expression of Snail, they should show the mRNA expression of RAC1B and TGFBR1 that the pathway was active. As Snail is regulated by other pathways.
In Figure 3, The authors could show the immunoblot for Snail1 it will be more convincing.
The authors could incorporate some images of the migration assays that were performed.
The authors can use CAGA Luciferase assay to show evidence that the pathway is activated by canonical Smad signaling. As p38 signaling has also been reported to activate snail1 expression.
There are recent reports stating that the TGFbeta Receptor I is cleaved and ICD is translocating to the nucleus to regulate Snail1 expression. The authors could comment on this findings in their discussion.
Author Response
Dear Editor:
This letter of submission is accompanied by our revised manuscript entitled:
RAC1B Suppresses TGF-b1-Dependent Cell Migration in Pancreatic Carcinoma Cells Through Inhibition of the TGF-b Type I Receptor ALK5
We have done our best to satisfy the reviewers’ concerns and have incorporated most of the requested changes into the revised version of our manuscript. The most important alterations to the original version have been highlighted in the “track changes” mode. Please also see below for a brief summary of the major changes made. We are confident that the reviewers’ critiques have substantially improved the quality of our manuscript and we are looking forward to its final acceptance in Cancers.
Faithfully yours,
Hendrik Ungefroren
General changes
1. All three reviewers refer to the SLUG data and since SLUG is an EMT-associated transcription factor asked for additional data on the role of RAC1B in epithelial-mesenchymal transition (EMT). Since we have in a previous publication extensively studied the role of RAC1B in TGF-b-induced EMT (see Ref. 3), we prefer not to include additional aspects of EMT in the present paper but rather focus on cell migration. For this reason, we decided to remove the SLUG data (Figure 2B) from the manuscript and replace them by data from reporter gene assays with the SMAD3-responsive reporter p(CAGA)12-luc (specifically requested by reviewer 1, Figure 2B in the revised version) and from proliferation assays (specifically requested by reviewer 3, Figure 2C in the revised version). These new data in panels B and C better match the SMAD3C activation data in Figure 2A since they provide functional proof of SMAD3, independent of cell migration.
2. The addition of a large number of new data in the revised version, particularly those with the Colo357 cell line, required a few alterations to the manuscript structure:
a) In the course of preparing the revision we have switched the order of Figures 1 and 3. We believe that this increases the fluency and readability of the paper since the expression data on ALK5 now precede all functional data (SMAD3-dependent activities in Figure 2 and migratory activities in Figures 3-5.
b) The new data with Colo357 cells have been included in Figures S2, S4, and S5.
c) The data in former Figure S2 have been moved to Figure 1 as panel A.
d) In Figure 2, the Slug qPCR data in panel B have been replaced by data from reporter gene assays with p(CAGA)12-luc, while a third panel, C, with data on growth inhibition have been added.
e) In Figure 5, panel A, the migration data have been replaced by Panc1-RAC1B-KO cells transiently transfected with kinase-dead ALK5 (specifically requested by reviewer 3).
f) The graph in Figure 6 has been moved to Figure 5B since the experiments in Figures 5 and 6 are thematically related (analysis of the ALK5 kinase function).
Review 1
The article is interesting, However the authors needs to address the comments below. The authors in Figure 2B show only the expression of Snail, they should show the mRNA expression of RAC1B and TGFBR1 that the pathway was active. As Snail is regulated by other pathways.
Response: We decided to remove the data on Snail2/Slug (Figure 2B) for the following reasons: 1) Regulation of TGF-b-induced EMT has been extensively characterized in Witte et al. 2) In response to requests from this reviewer and reviewer 3 we wanted to focus on SMAD(3)-dependent responses. The SLUG data have, therefore, been replaced by data on SMAD3 transcriptional activity (reporter gene assays with p(CAGA)12-luc) and growth inhibition (cell counting). Although SLUG regulation by TGF-b appears to be Smad-dependent, we were unable to find any entry in PubMed showing that specifically SMAD3 activation is involved here.
We have switched the order of Figures 3 and 1. This has the advantage that the subsections containing migration data now follow each other (Figures 3 through 5), making the Results section more coherent.
In Figure 3, The authors could show the immunoblot for Snail1 it will be more convincing. The authors could incorporate some images of the migration assays that were performed. The authors can use CAGA Luciferase assay to show evidence that the pathway is activated by canonical Smad signaling. As p38 signaling has also been reported to activate snail1 expression.
There are recent reports stating that the TGFbeta Receptor I is cleaved and ICD is translocating to the nucleus to regulate Snail1 expression. The authors could comment on this findings in their discussion.
Response: Since we decided to remove the SLUG data in Figure 2B (see above), immunoblot data are not applicable anymore. The CAGA luciferase assays in Panc1-RAC1B-KO cells have also been done and these data have replaced the SLUG qPCR data in Figure 2B.
The reports on p38 activation and cleaved receptor appear to refer to Snail1 rather than Snail2/Slug and we were unable to find equivalent sudies for Slug in PubMed. However, due to the removal of the SLUG data from our manuscript (see above), we would prefer not to comment on this finding.
Reviewer 2 Report
The study entitled “RAC1B Suppresses TGF-β1-Dependent Cell Migration in Pancreatic Tumor Cells Through Inhibition of the TGF- β Type 1 Receptor ALK5” by Ungefroren et al. further confirm that the small GTPase RAC1B represents an inhibitor of Transforming Growth Factor (TGF- β) signaling in pancreatic cells. They demonstrate RAC1B‘s inhibitory activity by comparing a siRNA that mediate a knockdown (KD) of RAC1B expression, with a CRISPR/Cas9-mediated knockout (KO) of RAC1B. They observe that both KD and KO of RAC1B increases the expression of the TGF- β type 1 receptor ALK5 and that this effect is more pronounced in CRISPR-KO cells. These data support the efficacy of RAC1B as an inhibitor of TGF- β signaling, and show potential strategies in cancers in which TGF- β has a pathogenic role.
The study is well organized even if the confirm of the results obtained on another PDAC cell line would have given it more worth.
Major points:
- When the authors analyze the role of RAC1B on SNAI2/SLUG they must show not only the increase of mRNA expression but also the protein levels. It is well-known that SNAI2/SLUG repress E-cadherin expression in EMT program therefore the authors must analyze its expression.
Minor points:
- The authors must show the immunoblot after KO and not only report as data not shown
- At the end of Introduction the authors report “....using Panc1 cells as the primary cellular system”, I suggest to remove this sentence because they used only Panc1 cell line.
- The authors should improve the figures:
· in Figure 1 they must show the bar graph relating migration reported in panel A;
· in Figure S1B and Figure 3 the authors must write more details on immunoblot shown as reported in Figure S2B
- There are many typographical errors:
· In the introduction: page 1, line 42 there is a round bracket randomly placed, I suggest to remove the sentence “and references therein”; page 2, line 49 there are a round bracket and colon randomly placed;
· In the results: page 3, line 96 TGF-B1-induced is repeated twice
· In the legend of figure 4, page 6, line 171, the word “significance” is randomly placed.
- The conclusions should be improved
Author Response
Dear Editor:
This letter of submission is accompanied by our revised manuscript entitled:
RAC1B Suppresses TGF-b1-Dependent Cell Migration in Pancreatic Carcinoma Cells Through Inhibition of the TGF-b Type I Receptor ALK5
We have done our best to satisfy the reviewers’ concerns and have incorporated most of the requested changes into the revised version of our manuscript. The most important alterations to the original version have been highlighted in the “track changes” mode. Please also see below for a brief summary of the major changes made. We are confident that the reviewers’ critiques have substantially improved the quality of our manuscript and we are looking forward to its final acceptance in Cancers.
Faithfully yours,
Hendrik Ungefroren
General changes
1. All three reviewers refer to the SLUG data and since SLUG is an EMT-associated transcription factor asked for additional data on the role of RAC1B in epithelial-mesenchymal transition (EMT). Since we have in a previous publication extensively studied the role of RAC1B in TGF-b-induced EMT (see Ref. 3), we prefer not to include additional aspects of EMT in the present paper but rather focus on cell migration. For this reason, we decided to remove the SLUG data (Figure 2B) from the manuscript and replace them by data from reporter gene assays with the SMAD3-responsive reporter p(CAGA)12-luc (specifically requested by reviewer 1, Figure 2B in the revised version) and from proliferation assays (specifically requested by reviewer 3, Figure 2C in the revised version). These new data in panels B and C better match the SMAD3C activation data in Figure 2A since they provide functional proof of SMAD3, independent of cell migration.
2. The addition of a large number of new data in the revised version, particularly those with the Colo357 cell line, required a few alterations to the manuscript structure:
a) In the course of preparing the revision we have switched the order of Figures 1 and 3. We believe that this increases the fluency and readability of the paper since the expression data on ALK5 now precede all functional data (SMAD3-dependent activities in Figure 2 and migratory activities in Figures 3-5.
b) The new data with Colo357 cells have been included in Figures S2, S4, and S5.
c) The data in former Figure S2 have been moved to Figure 1 as panel A.
d) In Figure 2, the Slug qPCR data in panel B have been replaced by data from reporter gene assays with p(CAGA)12-luc, while a third panel, C, with data on growth inhibition have been added.
e) In Figure 5, panel A, the migration data have been replaced by Panc1-RAC1B-KO cells transiently transfected with kinase-dead ALK5 (specifically requested by reviewer 3).
f) The graph in Figure 6 has been moved to Figure 5B since the experiments in Figures 5 and 6 are thematically related (analysis of the ALK5 kinase function).
Review 2
The study entitled “RAC1B Suppresses TGF-β1 Dependent Cell Migration in Pancreatic Tumor Cells Through Inhibition of the TGFβ Type 1 Receptor ALK5” by Ungefroren et al. further confirm that the small GTPase RAC1B represents an inhibitor of Transforming Growth Factor (TGFβ) signaling in pancreatic cells. They demonstrate RAC1B‘s inhibitory activity by comparing a siRNA that mediate a knockdown (KD) of RAC1B expression, with a CRISPR/Cas9mediated knockout (KO) of RAC1B. They observe that both KD and KO of RAC1B increasesthe expression of the TGFβ type 1 receptor ALK5 and that this effect is more pronounced in CRISPRKO cells. These data support the efficacyof RAC1B as an inhibitor of TGFβ signaling, and show potential strategies in cancers in which TGF- β has a pathogenic role. The study is well organized even if the confirm of the results obtained on another PDAC cell line would have given it more worth.
Major points:
When the authors analyze the role of RAC1B on SNAIL2/SLUG they must show not only the increase of mRNA expression but also the protein levels. It is well-known that SNAIL2/SLUG repress E-cadherin expression in EMT program therefore the authors must analyze its expression.
Response: As mentioned above, we would prefer to remove the data on SLUG from the manuscript for the reasons stated in our response to comment #1 of Reviewer 1. TGF-b1 regulation of E-cadherin protein expression in RAC1B knockdown cells has been analyzed previously by immunoblotting (see Ref. 3).
Minor points:
The authors must show the immunoblot after KO and not only report as data not shown
Response: An immunoblot of RAC1B and RAC1 in KO cells has been added to Figure S1A.
At the end of Introduction the authors report “....using Panc1 cells as the primary cellular system”, I suggest to remove this sentence because they used only Panc1 cell line.
Response: This sentence should remain because we added data from another cell line (Colo357) during the revision (in response to requests from Reviewer 3 and the Academic Editor). Please see Figures S2, S4, and S5.
The authors should improve the figures:
in Figure 1 they must show the bar graph relating migration reported in panel A.
Response: Done. Please note that Figure 1 has become Figure 3 in the revised version.
In Figure S1B and Figure 3 the authors must write more details on immunoblot shown as reported in Figure S2B.
Response: Done for Figure S1B and former Figure 3 (Figure 1 in the revised version).
There are many typographical errors:
In the introduction: page 1, line 42 there is a round bracket randomly placed, I suggest to remove the sentence “and references therein”; page 2, line 49 there are a round bracket and colon randomly placed;
Response: All rectified.
In the results: page 3, line 96 TGF-B1-induced is repeated twice
In the legend of figure 4, page 6, line 171, the word “significance” is randomly placed.
Response: Rectified.
The conclusions should be improved.
Response: Rectified.
Reviewer 3 Report
In the present manuscript, the authors performed experiments showing that RAC1B, a GTPase encoded by the RAC1 gene, inhibits TGF B-dependent cell migration. This phenomenon is accomplished through inhibition of SMAD3, p38MAPK and ERK1/2 MAP kinase functions and is accompanied by downregulation of the TGF beta type I receptor namely ALK5. While a KD based experimental approach has already been performed in the past with similar results, in this study, the authors present a CRISPR/Cas9 mediated KO of RAC1B in pancreatic cancer cells (Panc1 cell line).
Major concerns
The experiments are well performed and conceived, with appropriate controls. An important issue regards the statistical analysis of data. Information about the performed statistical analysis (which kind of test has been run) is missing, both in figure legends and in the M&M section. In Figure 3B, it would be interesting to see if the value of KO in TGFB1 minus sample (KO white bar) is statistically different from the wt in the presence of TGFb1 (V black bar). The same can be said for experiment 3B.
Although I see the point of using dominant negative ALK 5 mutant to analyze RAC1b role in ALK5 dependent pathway (experiments shown in figure 5), the observation that these cells do not migrate at all makes them not useful to assess RAC1B role in migration. As suggested by authors, the overexpression of KO cells with ALK5 dominant negative mutants should be performed instead. The authors should perform RAC1silencing in Panc1-ALK5T204D cell line and check the migration upon TGF induction. If the author hypothesis is correct no further increase in migration properties should be observed in KO (or KD) cells when the mutant T204D is expressed.
Given the interplay between cytoskeleton dynamics and cellular migration, I think it would be interesting to add a phalloidin staining of KD cells to visualize actin cytoskeleton, or immunofluorescence analysis of proteins that mediate cytoskeleton re-organization, and either cell-cell or cell-substrate adhesion.
The authors observe that in the absence of RAC1B expression cells display EMT as assessed by qRT-PCR. To give further strength to this observation, the authors should provide a western blot analysis showing the increase of molecular players of EMT as well as the proliferation rate of these cells in the presence of TGF 1.
Does this data apply to other cancers or is something highly specific of lung cancer? Experiments showing the role of RAC1B in at least another cell context would increase the interest of the readership.
Author Response
Dear Editor:
This letter of submission is accompanied by our revised manuscript entitled:
RAC1B Suppresses TGF-b1-Dependent Cell Migration in Pancreatic Carcinoma Cells Through Inhibition of the TGF-b Type I Receptor ALK5
We have done our best to satisfy the reviewers’ concerns and have incorporated most of the requested changes into the revised version of our manuscript. The most important alterations to the original version have been highlighted in the “track changes” mode. Please also see below for a brief summary of the major changes made. We are confident that the reviewers’ critiques have substantially improved the quality of our manuscript and we are looking forward to its final acceptance in Cancers.
Faithfully yours,
Hendrik Ungefroren
General changes
1. All three reviewers refer to the SLUG data and since SLUG is an EMT-associated transcription factor asked for additional data on the role of RAC1B in epithelial-mesenchymal transition (EMT). Since we have in a previous publication extensively studied the role of RAC1B in TGF-b-induced EMT (see Ref. 3), we prefer not to include additional aspects of EMT in the present paper but rather focus on cell migration. For this reason, we decided to remove the SLUG data (Figure 2B) from the manuscript and replace them by data from reporter gene assays with the SMAD3-responsive reporter p(CAGA)12-luc (specifically requested by reviewer 1, Figure 2B in the revised version) and from proliferation assays (specifically requested by reviewer 3, Figure 2C in the revised version). These new data in panels B and C better match the SMAD3C activation data in Figure 2A since they provide functional proof of SMAD3, independent of cell migration.
2. The addition of a large number of new data in the revised version, particularly those with the Colo357 cell line, required a few alterations to the manuscript structure:
a) In the course of preparing the revision we have switched the order of Figures 1 and 3. We believe that this increases the fluency and readability of the paper since the expression data on ALK5 now precede all functional data (SMAD3-dependent activities in Figure 2 and migratory activities in Figures 3-5.
b) The new data with Colo357 cells have been included in Figures S2, S4, and S5.
c) The data in former Figure S2 have been moved to Figure 1 as panel A.
d) In Figure 2, the Slug qPCR data in panel B have been replaced by data from reporter gene assays with p(CAGA)12-luc, while a third panel, C, with data on growth inhibition have been added.
e) In Figure 5, panel A, the migration data have been replaced by Panc1-RAC1B-KO cells transiently transfected with kinase-dead ALK5 (specifically requested by reviewer 3).
f) The graph in Figure 6 has been moved to Figure 5B since the experiments in Figures 5 and 6 are thematically related (analysis of the ALK5 kinase function).
Review 3
In the present manuscript, the authors performed experiments showing that RAC1B, a GTPase encoded by the RAC1 gene, inhibits TGF B-dependent cell migration. This phenomenon is accomplished through inhibition of SMAD3, p38MAPK and ERK1/2 MAP kinase functions and is accompanied by downregulation of the TGF beta type I receptor namely ALK5. While a KD based experimental approach has already been performed in the past with similar results, in this study, the authors present a CRISPR/Cas9 mediated KO of RAC1B in pancreatic cancer cells (Panc1 cell line).
Major concerns
The experiments are well performed and conceived, with appropriate controls. An important issue regards the statistical analysis of data. Information about the performed statistical analysis (which kind of test has been run) is missing, both in figure legends and in the M&M section.
Response: This information is given under section 5.8. (5.9. in the revised version): “Statistical significance was calculated using the unpaired two-tailed Student’s t test. Results were considered significant at p<0.05 (*).”. However, we have now indicated this also in the main text and figure legends where appropriate.
In Figure 3B, it would be interesting to see if the value of KO in TGFB1 minus sample (KO white bar) is statistically different from the wt in the presence of TGFb1 (V black bar). The same can be said for experiment 3B.
Response: While in panel A of Figure 3 (Figure 1B, left panel, in the revised version), the difference between both values missed statistical significance (p=0.0636), in panel B (Figure 1B, right panel, in the revised version) the difference is indeed significant (p=0.0042). This has now been indicated in the revised panels.
Although I see the point of using dominant negative ALK 5 mutant to analyze RAC1b role in ALK5 dependent pathway (experiments shown in figure 5), the observation that these cells do not migrate at all makes them not useful to assess RAC1B role in migration. As suggested by authors, the overexpression of KO cells with ALK5 dominant negative mutants should be performed instead.
Response: This is a good suggestion. However, it should be mentioned that the strong inhibitory effect of the ALK5-KD mutant results from the fact that ALL cells of the population express this mutant (genetically homogenous population) and that a fraction of the “basal” migration in the corresponding vector control cells is indeed TGF-b-dependent migration due to the high degree of autocrine TGF-b stimulation. Nevertheless, we have performed transient overexpression of ALK5-KR in Panc1-RAC1B-KO cells and data are now shown in Figure 5A. However, as expected, the effect of transient expression of ALK5-KR was only partial for two reasons: 1) in contrast to the stable transductants with 100% of cells expressing the ALK5 mutant, transient transfection results in only a fraction of cells expressing it, and 2) due to the higher endogenous ALK5 expression in the RAC1B-KO, the levels of ectopically expressed ALK5-KR achieved by transient transfection might not be able to compete out all endogenous ALK5. In a parallel approach, we have treated Colo357-RAC1B-KD cells with the ALK5 kinase inhibitor SB431542 and have obtained results that are consistent with those from the dominant-negative inhibition approach in Panc1 cells (see Figure S5).
The authors should perform RAC1 silencing in Panc1-ALK5T204D cell line and check the migration upon TGF induction. If the author hypothesis is correct no further increase in migration properties should be observed in KO (or KD) cells when the mutant T204D is expressed.
Response: Again, we appreciate this suggestion of the reviewer. However, we believe that the novel data with Panc1-RAC1B-KO cells and the dominant-negative mutant (Figure 5A) plus the Colo357-RAC1B-KD cells and SB431542 (Figure S5) are sufficient to demonstrate a crucial role for the ALK5 kinase activity in RAC1B inhibition-mediated increase in cell migration. For the same reason, we have moved the migration data of Panc1-ALK5-TD cells from former Figure 6 to Figure 5 as panel B, since these data are conceptually associated with the role of ALK5 kinase activity.
Given the interplay between cytoskeleton dynamics and cellular migration, I think it would be interesting to add a phalloidin staining of KD cells to visualize actin cytoskeleton, or immunofluorescence analysis of proteins that mediate cytoskeleton re-organization, and either cell-cell or cell-substrate adhesion.
Response: This is an interesting suggestion from a cell biological perspective. However, since these processes are not directly related to ALK5 function, we feel that these experiments are beyond the scope of our study. We would therefore prefer, not to add these data.
The authors observe that in the absence of RAC1B expression cells display EMT as assessed by qRT-PCR. To give further strength to this observation, the authors should provide a western blot analysis showing the increase of molecular players of EMT as well as the proliferation rate of these cells in the presence of TGF-b1.
Response: The negative control of EMT by RAC1B has been extensively analyzed in Witte et al. 2017 (Ref. 3, sell also response to comment #1 of reviewer 1). Our study from 2017 also contains immunoblots for E-cadherin and SNAIL. In the present study, we would, therefore, prefer to focus on migration rather than on additional EMT markers (please also see our response to comment #1 of reviewer 1).
As suggested, we have analyzed TGF-b-regulated proliferation in Panc1-RAC1B-KO and vector control cells by cell counting experiments and have found that the growth inhibitory effect of TGF-b1 is much more pronounced in the KO cells after 24 and 48 hours (see new Figure 2C).
Does this data apply to other cancers or is something highly specific of lung cancer? Experiments showing the role of RAC1B in at least another cell context would increase the interest of the readership.
Response: Negative regulation of TGF-b-induced cell migration by RAC1B has been shown for other pancreatic cells: HPDE cells and Colo357 cells (Ungefroren et al. 2014), breast cancer cells (Melzer et al. 2017) and prostate carcinoma cells (HU, unpublished observation). With regard to the functional role of ALK5, we have performed double-knockdown of RAC1B and ALK5 as well as pharmacologic inhibition of the ALK5 kinase in another TGF-b-sensitive PDAC-derived cell line, Colo357. The results of these assays are consistent with those in Panc1 and are shown in Figures S2, S4, and S5 in the revised version.
Round 2
Reviewer 2 Report
In my opinion the manuscript in this form is suitable for publication in Cancers